# Nitrate Modulates Lateral Root Formation by Regulating the Auxin Response and Transport in Rice

**DOI:** 10.3390/genes12060850

**Published:** 2021-06-01

**Authors:** Bobo Wang, Xiuli Zhu, Xiaoli Guo, Xuejiao Qi, Fan Feng, Yali Zhang, Quanzhi Zhao, Dan Han, Huwei Sun

**Affiliations:** 1Key Laboratory of Rice Biology in Henan Province, Collaborative Innovation Center of Henan Grain Crops, Henan Agricultural University, Zhengzhou 450002, China; wb1991xz@163.com (B.W.); zhuxiuli03@163.com (X.Z.); guojingli188@126.com (X.G.); qixuejiao1218@163.com (X.Q.); fengfanqqq@126.com (F.F.); qzzhaoh@henau.edu.cn (Q.Z.); 2Key Laboratory of Plant Nutrition and Fertilization in Low-Middle Reaches of the Yangtze River, Ministry of Agriculture, Nanjing Agricultural University, Nanjing 210095, China; ylzhang@njau.edu.cn; 3College of Tobacco Science, Henan Agricultural University, Zhengzhou 450002, China

**Keywords:** ammonium, auxin, lateral root (LR), nitrate, rice, strigolactones (SLs)

## Abstract

Nitrate (${\mathrm{NO}}_{3}^{-}$) plays a pivotal role in stimulating lateral root (LR) formation and growth in plants. However, the role of ${\mathrm{NO}}_{3}^{-}$ in modulating rice LR formation and the signalling pathways involved in this process remain unclear. Phenotypic and genetic analyses of rice were used to explore the role of strigolactones (SLs) and auxin in ${\mathrm{NO}}_{3}^{-}$-modulated LR formation in rice. Compared with ammonium (${\mathrm{NH}}_{4}^{+}$), ${\mathrm{NO}}_{3}^{-}$ stimulated LR initiation due to higher short-term root IAA levels. However, this stimulation vanished after 7 d, and the LR density was reduced, in parallel with the auxin levels. Application of the exogenous auxin α-naphthylacetic acid to ${\mathrm{NH}}_{4}^{+}$-treated rice plants promoted LR initiation to levels similar to those under ${\mathrm{NO}}_{3}^{-}$ at 7 d; conversely, the application of the SL analogue GR24 to ${\mathrm{NH}}_{4}^{+}$-treated rice inhibited LR initiation to levels similar to those under ${\mathrm{NO}}_{3}^{-}\;$supply by reducing the root auxin levels at 10 d. *D10* and *D14* mutations caused loss of sensitivity of the LR formation response to ${\mathrm{NO}}_{3}^{-}$. The application of ${\mathrm{NO}}_{3}^{-}\;$and GR24 downregulated the transcription of *PIN-FORMED 2*
*(PIN2)*, an auxin efflux carrier in roots. LR number and density in *pin2* mutant lines were insensitive to ${\mathrm{NO}}_{3}^{-}$ treatment. These results indicate that ${\mathrm{NO}}_{3}^{-}\;\mathrm{modulates}$ LR formation by affecting the auxin response and transport in rice, with the involvement of SLs.

## 1. Introduction

Plants have various mechanisms to adapt to nutrient supply conditions, especially plastic root development [1,2,3,4,5,6,7]. Lateral roots (LRs) are generally more sensitive to nutrient conditions than that are primary/adventitious roots in plants [8,9]. The LRs develop from founder cells in the pericycle, the outermost layer of the vascular cylinder (stele) of the roots [10].

Nitrogen (N) is an essential macronutrient for plant growth and crop productivity. Changes in N supplied in the nutrient medium induce plasticity in LR initiation and elongation [5,11,12,13,14]. A striking example of plasticity in LR development is seen in the response of *Arabidopsis* to localised ${\mathrm{NO}}_{3}^{-}$ treatment via the stimulation of LR elongation. Studies of an *Arabidopsis* nitrate reductase double mutant suggested that the local stimulation of LR elongation is a consequence of the ${\mathrm{NO}}_{3}^{-}$ ion acting as a signal rather than a nutrient. Nitrate transporters, transcription factors, and micro-RNAs are involved in ${\mathrm{NO}}_{3}^{-}$-modulated LR growth and development [14,15,16,17,18,19]. LR growth is regulated by both environmental conditions and intrinsic developmental regulators, such as plant hormones [20]. Auxin plays a dominant role in the specification of founder cells that give rise to LR initiation and the later stages of LR development and is involved in ${\mathrm{NO}}_{3}^{-}$-modulated LR growth [10,13,14,20,21]. Localised ${\mathrm{NO}}_{3}^{-}$ supply does not stimulate LR elongation in *axr4*, an auxin-insensitive mutant, which suggests that ${\mathrm{NO}}_{3}^{-}$ regulates LR growth via auxin signalling pathways [21]. NRT1.1 is a key component of ${\mathrm{NO}}_{3}^{-}$-sensing system that enables the plant to detect and exploit ${\mathrm{NO}}_{3}^{-}$ [22]. The ${\mathrm{NO}}_{3}^{-}$ and auxin signalling pathways are also linked by their effect on auxin transport via *AtNRT1.1* (*CHL1/NPF6.3*) [23]. Local high levels of ${\mathrm{NO}}_{3}^{-}$ promoted *Arabidopsis* LR development as a result of auxin accumulation in the LR primordia and tip [24]. However, inconsistent results have been reported in maize [25], although localised ${\mathrm{NO}}_{3}^{-}$-induced LR elongation has been observed, ${\mathrm{NO}}_{3}^{-}$-fed compartments have lower auxin levels compared with ${\mathrm{NO}}_{3}^{-}$-free compartments, and localised ${\mathrm{NO}}_{3}^{-}$ supply inhibits auxin transport from shoot to root. A positive effect of low ${\mathrm{NO}}_{3}^{-}$ on *Arabidopsis* LR formation required more auxin accumulation in LR primordia [20], consistent with the result in maize [26]; however, LR formation in rice was inhibited by low ${\mathrm{NO}}_{3}^{-}$, which was closely linked to lower auxin contents. The role of auxin in ${\mathrm{NO}}_{3}^{-}$-regulated LR growth remains unclear.

Strigolactones (SLs) are phytohormones involved in the growth and formation of LR in several plant species [27,28,29,30]. Compared with the wild-type (WT), *Arabidopsis* with mutations associated with SL synthesis and signalling had higher LR densities [27]. However, there was no difference in LR density between WT and *d* mutants in rice [30]. Application of GR24 decreased the LR density in both *Arabidopsis* and rice [27,30]. SLs are also involved in ${\mathrm{NO}}_{3}^{-}$-regulated root elongation by modulating *PIN1b* gene expression [7]. Therefore, the mechanisms by which SLs regulate LR growth in response to ${\mathrm{NO}}_{3}^{-}$ supply are more complex and require further investigation.

Studies of LR growth in response to ${\mathrm{NO}}_{3}^{-}$ have focused on the upland model plant *Arabidopsis*, and research in other plants, especially crop plants, is needed. Rice (*Oryza sativa* L.) is a major staple food globally, and ${\mathrm{NH}}_{4}^{+}\;\mathrm{provides}$ the main source of N for rice in paddy soil [31]. Interestingly, it has been predicted that 40% of the N acquired by rice roots is ${\mathrm{NO}}_{3}^{-}$ due to nitrification occurring at the root surface, even in flooded conditions [32,33]. Increasing numbers of Chinese farmers are practicing intermittent flooding during rice cultivation, which increases ${\mathrm{NO}}_{3}^{-}$ within the soil horizon. Although ${\mathrm{NO}}_{3}^{-}$ plays a pivotal role in regulating root architecture by stimulating the initiation and elongation of LRs, the role of ${\mathrm{NO}}_{3}^{-}$ in modulating LR growth in rice and the signalling pathways involved in this process remain unclear. Therefore, to evaluate the mechanisms of ${\mathrm{NO}}_{3}^{-}$-modulated LR formation in rice, we compared the time course of LR formation, auxin content, and *DR5::GUS* activity of rice in response to ${\mathrm{NO}}_{3}^{-}$ and ${\mathrm{NH}}_{4}^{+}$.

## 2. Results

### 2.1. Nitrate Regulated LR Formation in Rice

Compared with ${\mathrm{NH}}_{4}^{+}$ conditions, the number of LRs in the seminal root increased under ${\mathrm{NO}}_{3}^{-}$ treatment within 7 d (Figure 1A,B). However, the LR number was lower under ${\mathrm{NO}}_{3}^{-}$ than ${\mathrm{NH}}_{4}^{+}$ treatment after 10 d (Figure 1B). There was no difference in LR density between ${\mathrm{NH}}_{4}^{+}$ and ${\mathrm{NO}}_{3}^{-}$ conditions before 7 d. Surprisingly, LR density was lower under ${\mathrm{NO}}_{3}^{-}$ than ${\mathrm{NH}}_{4}^{+}$ treatment after 8 d (Figure 1C,D). These results suggest that ${\mathrm{NO}}_{3}^{-}$ supply stimulates LR formation for a short period (within 7 d), but this stimulatory effect disappears after 7 d.

### 2.2. Auxin Is Involved in ${NO}_{3}^{-}$-Modulated LR Formation

Abundant evidence suggests that auxin has a close relationship with LR development [3,10,13,21]. To understand temporal changes in auxin-responsive genes to ${\mathrm{NO}}_{3}^{-}$, *DR5**::GUS* and the expression of *AUXIN RESPONSE FACTOR 1* (*ARF**1*) in roots under ${\mathrm{NH}}_{4}^{+}$, ${\mathrm{NO}}_{3}^{-},$ and ${\mathrm{NH}}_{4}^{+}$ plus α-naphthylacetic acid (NAA) treatments were analysed from 0 to 12 h (Figure 2). The expression of *DR5::GUS* in roots was induced by ${\mathrm{NO}}_{3}^{-}$ and ${\mathrm{NH}}_{4}^{+}$ plus NAA over the entire experiment compared with ${\mathrm{NH}}_{4}^{+}$ nutrition (Figure 2A). Compared with ${\mathrm{NH}}_{4}^{+}$, the expression of *OsARF**1* was upregulated by both ${\mathrm{NO}}_{3}^{-}$ and ${\mathrm{NH}}_{4}^{+}$ plus NAA (Figure 2B).

To assess the roles of auxin in ${\mathrm{NO}}_{3}^{-}$-induced LR formation in rice, we examined the LR number in response to exogenous application of NAA under ${\mathrm{NH}}_{4}^{+}$ at 7 d. Application of ${\mathrm{NH}}_{4}^{+}$ plus NAA significantly increased the LR number at 7 d, to the same level as that under ${\mathrm{NO}}_{3}^{-}$ conditions (Figure 2C).

To determine whether *ARF1* is involved in ${\mathrm{NO}}_{3}^{-}$-promoted rice LR formation in the short term, we used *arf1* mutant. The T-DNA insertion mutant of *arf1* is shown in Appendix A. Compared with WT plants (DJ), LR number and density were decreased in the *arf1* mutant under both ${\mathrm{NH}}_{4}^{+}$ and ${\mathrm{NO}}_{3}^{-}$ conditions (Figure 3), indicating that *ARF1* is involved in ${\mathrm{NO}}_{3}^{-}$-induced LR formation in rice.

### 2.3. SLs Are Also Involved in ${NO}_{3}^{-}$-Modulated LR Formation

Compared with WT plants, the root morphology of *d10* (SL biosynthesis mutant) and *d**14* (SL-responsive mutant) plants, including LR density, was less responsive to ${\mathrm{NO}}_{3}^{-}$ (Figure 4A,B). For example, the LR density under the two treatments was similar between *d* mutants and ${\mathrm{NH}}_{4}^{+}$-treated WT plants at 10 d. Interestingly, the LR density of *d* mutants was less responsive to ${\mathrm{NO}}_{3}^{-}$ supply, ultimately resulting in a greater LR density, compared with WT plants regardless of the treatment at 10 d (Figure 4A). These results suggest that SLs are involved in ${\mathrm{NO}}_{3}^{-}$-modulated LR formation in rice.

Based on the LR density of *d* mutants in response to ${\mathrm{NO}}_{3}^{-},\;$we speculated that the IAA content is higher in the roots of *d* mutants at 10 d (Figure 4B). Compared with ${\mathrm{NH}}_{4}^{+}$, ${\mathrm{NO}}_{3}^{-}$ treatment reduced the IAA levels in the roots of WT plants. IAA levels were similar between WT and *d* mutants under ${\mathrm{NH}}_{4}^{+}$ conditions, but were higher in *d* mutants than WT plants under ${\mathrm{NO}}_{3}^{-}$ conditions (Figure 4B). We examined whether exogenous application of the SL analogue GR24 affects the IAA levels and LR formation (Figure 5). The application of GR24 in ${\mathrm{NH}}_{4}^{+}$-treated rice reduced *DR5::GUS* expression and IAA levels in roots (Figure 5A,B), and inhibited LR formation to levels similar to those under ${\mathrm{NO}}_{3}^{-}$ at 10 d (Figure 5C). Conversely, treatment with NAA plus ${\mathrm{NO}}_{3}^{-}$ significantly increased the LR density to the same level as that under ${\mathrm{NH}}_{4}^{+}$ at 10 d (Figure 5D). These results indicate that ${\mathrm{NO}}_{3}^{-}$ inhibited LR formation, probably by decreasing auxin levels in roots in the long term, and SLs may be involved in this process.

### 2.4. OsPIN2 Is Involved in ${NO}_{3}^{-}$-Modulated Auxin Levels and LR Formation in Rice

A previous study showed that SLs regulate LR formation by inhibiting auxin transport, with involvement of PIN proteins [30]. In this study, compared with ${\mathrm{NH}}_{4}^{+}$, ${\mathrm{NO}}_{3}^{-}$ treatment downregulated the expression of *PIN2* and *proPIN2::GUS* at 10 d (Figure 6A,C). *PIN2* expression in roots was significantly higher in the *d**14* mutant than WT plants under both treatments (Figure 6B). *PIN2* expression was downregulated by ${\mathrm{NH}}_{4}^{+}$ plus GR24 compared with ${\mathrm{NH}}_{4}^{+}$ treatment at 10 d (Figure 6C). These results suggest that SLs participate in the ${\mathrm{NO}}_{3}^{-}$-induced inhibition of *PIN2* transcription gene in rice.

To determine the functions of *PIN2* in ${\mathrm{NO}}_{3}^{-}$-modulated LR formation, we assessed the LR number and density in *pin2* mutants in response to ${\mathrm{NO}}_{3}^{-}$ at 10 d (Appendix A; Figure 7). Compared with WT plants, the two *pin2* mutant lines exhibited less responsiveness of the LR number and density to ${\mathrm{NO}}_{3}^{-}$ and fewer and less dense LRs under the two treatments at 10 d (Figure 7B,C). This implies that *OsPIN2* is also involved in rice LR formation modulated by ${\mathrm{NO}}_{3}^{-}$ application long-term.

## 3. Discussion

Nitrogen is a major plant nutrient, and crops strongly depend on fertilization programs, affecting environmental quality. The identification of crop cultivars with more efficient nutrient acquisition continues to be a priority for plant scientists [34,35]. LRs are crucial for the detection and uptake of N in plants [24,36]. Nitrate triggers several molecular and physiological events, including LR growth, leading to the overall response of the plant to its availability [11,12,37]. Several molecular components of ${\mathrm{NO}}_{3}^{-}$-regulated LR growth have been identified, mostly in the model plant *Arabidopsis thaliana*. However, the role of ${\mathrm{NO}}_{3}^{-}$ in modulating LR formation and growth in rice and the signalling pathways involved in this process remain unclear. In this study, we found that ${\mathrm{NO}}_{3}^{-}$-induced LR growth depends on the auxin response and transport in roots, with the involvement of SLs.

Studies of an *Arabidopsis* nitrate reductase double mutant suggested that the local stimulation of LR elongation was a consequence of the ${\mathrm{NO}}_{3}^{-}$ ion acting as a signal, rather than a nutrient. The *AtANR1* and *AtNRT1.1* genes, which encode a transcription factor and a dual ${\mathrm{NO}}_{3}^{-}$ transporter, respectively, were proposed to consecutively regulate the stimulatory effect of ${\mathrm{NO}}_{3}^{-}$ on LR elongation [11,21,23]. In rice, LR formation was less sensitive to localised ${\mathrm{NO}}_{3}^{-}$ supply in *osnar2.1* mutants than WT plants, suggesting that *OsNAR2.1* is involved in a ${\mathrm{NO}}_{3}^{-}$ signalling pathway that modulates LR formation [15]. Here, we also found that ${\mathrm{NO}}_{3}^{-}$ induced LR formation, probably via its signalling pathway. As shown in Figure 1, ${\mathrm{NO}}_{3}^{-}$ may induce LR formation by triggering systemic signals that influence LR growth compared with ${\mathrm{NH}}_{4}^{+}$.

Localised ${\mathrm{NO}}_{3}^{-}$ supply did not stimulate LR elongation in the auxin-insensitive mutant, which suggests that ${\mathrm{NO}}_{3}^{-}$ regulates LR growth via auxin signalling pathways [21]. To illustrate the mechanism of the nitrate-specific effects on rice LR formation within 7 d, the expression of *ARF1* (auxin-responsive gene) and *DR5::GUS* was evaluated in response to two N forms (Figure 2). After 3 h of treatment, *ARF**1* expression in rice roots was higher under ${\mathrm{NO}}_{3}^{-}$ than under ${\mathrm{NH}}_{4}^{+}$, which coincided with the higher *DR5::GUS* expression in the LR zone under$\;{\mathrm{NO}}_{3}^{-}$ treatment. ${\mathrm{NO}}_{3}^{-}$ enhanced *ARF1* expression within hours, suggesting that auxin triggers a systemic signal to participate in ${\mathrm{NO}}_{3}^{-}$-induced LR formation in rice. Exogenous application of NAA under ${\mathrm{NH}}_{4}^{+}$ supply restored the LR number to a level similar to that under ${\mathrm{NO}}_{3}^{-}$ supply within 7 d (Figure 2), and the LR number and density were lower in the *arf1* mutant relative to WT plants at 7 d (Figure 3B,C), which further demonstrates that auxin participates in specific ${\mathrm{NO}}_{3}^{-}$-induced LR formation.

SLs have been suggested to modulate auxin transport in the regulation of root growth [27,30]. In *Arabidopsis*, SLs modulated local auxin levels, and the net result of the SL action depended on the auxin status of the plant [27]. In rice, GR24 application markedly reduced auxin transport to levels equivalent to those under N-deficient conditions, which in turn reduced the LR density [30]. A previous study showed that ${\mathrm{NO}}_{3}^{-}$ application enhanced SL signalling from 7 d in rice [7]. The SL levels in root exudates are regulated by N stress [30]. In this study (Figure 5), application of GR24 to rice plants under ${\mathrm{NH}}_{4}^{+}$ treatment inhibited LR initiation to the same levels as those under ${\mathrm{NO}}_{3}^{-}$ treatment by reducing IAA levels in roots at 10 d. Conversely, compared with ${\mathrm{NO}}_{3}^{-}$ conditions, NAA treatment of ${\mathrm{NO}}_{3}^{-}$-treated rice prevented the downtrend in LR initiation to the same levels as those in ${\mathrm{NH}}_{4}^{+}$-treated rice plants at 10 d (Figure 5). This indicated that the ${\mathrm{NO}}_{3}^{-}$ supply increased SL production after 7 d and inhibited LR formation by decreasing auxin levels in the LR region, consistent with the previous report [7,30]. This suggests that SLs are involved in ${\mathrm{NO}}_{3}^{-}$-inhibited LR formation by reducing auxin transport in roots in the long term.

The influence of SLs on auxin transport is mediated by *PIN* expression [3,7,27,30]. For example, SLs increased the rate of PIN1 removal from the plasma membrane and altered the polarization of PIN2 in the plasma membrane in *Arabidopsis* [38]. Similarly, relative *PIN* expression in rice roots was significantly decreased under LN conditions, after GR24 application [30]. SLs participated in ${\mathrm{NO}}_{3}^{-}$-induced rice root elongation by modulating *PIN1b* transcription [7]. In this study, *PIN2* expression was inhibited by ${\mathrm{NO}}_{3}^{-}$ treatment long-term (Figure 6A,C), suggesting that *PIN2* is involved in ${\mathrm{NO}}_{3}^{-}$-modulated auxin polar transport to play an important role in LR development. Compared with ${\mathrm{NH}}_{4}^{+}$ treatment, *PIN2* expression was downregulated under ${\mathrm{NO}}_{3}^{-}$ or ${\mathrm{NH}}_{4}^{+}$ plus GR24 treatment at 10 d (Figure 6). Furthermore, the expression of *PIN2* was significantly upregulated in the *d**14* mutant compared with the WT (Figure 6B). These results suggest that ${\mathrm{NO}}_{3}^{-}$ inhibits auxin transport by regulating *PIN2* expression in roots, with the involvement of SLs. Compared with WT plants, mutations in *D* genes that eliminate the inhibition of SLs on auxin transport led to higher auxin levels in the LR region and no response of LR formation to ${\mathrm{NO}}_{3}^{-}$ relative to ${\mathrm{NH}}_{4}^{+}$ (Figure 4). The lower LR number and density were recorded in the *pin2* mutants relative to WT plants under both ${\mathrm{NH}}_{4}^{+}$ and ${\mathrm{NO}}_{3}^{-}$ supplies (Figure 7). These results further demonstrate that the effect of LR formation regulated by ${\mathrm{NO}}_{3}^{-}$ depends on auxin response and transport in roots.

## 4. Materials and Methods

### 4.1. Plant Materials

The *d10* (SL biosynthesis mutant) and *d14* (SL-responsive mutant) were Shiokari ecotype [30], *arf1* mutant was Dongjin (DJ) ecotype, and CRISPR-edited *PIN**2* knockout mutant lines (*pin2*) were Nipponbare ecotype. The *arf1* was obtained from Kyung Hee University, Korea (Appendix A).

### 4.2. Plant Growth

Plants were grown in a greenhouse under natural light at day/night temperatures of 30 °C/18 °C. Germinated seeds of uniform size were transplanted into holes in a PCR tube rack for 14 d. PCR tubes receiving nitrogen treatments were filled with 1.25 (NH_4_)_2_SO_4_ and/or Ca(NO_3_)_2_. Other chemical compositions of International Rice Research Institute (IRRI) nutrient solution were (mM): 1.25 (NH_4_)_2_SO_4_ and/or Ca(NO_3_)_2_, 0.3 KH_2_PO_4_, 0.35 K_2_SO_4_, 1.0 CaCl_2_, 1.0 MgSO_4_·7H_2_O, 0.5 Na_2_SiO_3_; and (µM) 9.0 MnCl_2_, 0.39 (NH_4_)_6_Mo_7_O_24_, 20.0 H_3_BO_3_, 0.77 ZnSO_4_, and 0.32 CuSO_4_ (pH 5.5) as previously described [30].

The treatments applied were as follows: 10 nM 1-naphthylacetic acid (NAA), 2.5 µM GR24 (an SL analogue) [30,39].

### 4.3. Root System Architecture

The fibrous root system of rice includes seminal root, adventitious roots, and lateral root (LR) grown from seminal and adventitious roots. The preliminary experiments suggested that the response of LRs on seminal root to two N forms was similar to that on adventitious roots. Therefore, the numbers of LRs on SRs were chosen to evaluate the effects of ${\mathrm{NH}}_{4}^{+}$ and ${\mathrm{NO}}_{3}^{-}$ on LR growth. LRs were enumerated visually. The LR density was calculated as LR number divided by the length of the SR.

### 4.4. Determination of IAA Content

The plant tissues were ground with quartz sand and butylated hydroxytoluene (BHT) in liquid N2 and lixiviated in 80% methanol (20 mL) for 12 h. The extracted fluid was collected and concentrated by a rotary evaporator to 10 mL at 40 mL, and then the concentrated fluid was extracted with petroleum ether of the same volume. Underlayer liquid was adjusted to pH 8.5 and added 0.2 g polyvinylpyrrolidone (PVP) then vibrated for 30 min, and then filtered through a 0.45 μm filter. The cartridge was initially washed with 0.1 M acetic acid, eluted with 4 mL of a mixture of 25% (*v*/*v*) methanol and 0.1 M acetic acid, and eventually with 70% (*v*/*v*) methanol only. After vacuum evaporation, the purified samples were metered volume to 1 mL with mobile phase and then loaded on a reverse-phase HPLC column. Standard auxin samples were from Sigma-Aldrich (St. Louis, MO, USA), and chromatographic conditions were described as: Waters 600–2487; Hibar column RT 250 mm × 4.6 mm; Purospher STAR RP-18 (5 μm); column temperature 45 °C; fluid phase: methanol: 1% acetic acid (*v*/*v*, 40/60), isocratic elution; fluid rate: 0.6 mL min^−1^; UV detector, *l* = 269 nm; injection volume 20 μL. A 0.22 μm filter was used for filtration of both the buffer and the samples before HPLC analysis as previously described [40].

To assess auxin distribution, rice plants were transformed with the *pDR5::GUS* constructs using *Agrobacterium tumefaciens* (strain EHA105). *DR5::GUS*, a specific reporter that contains seven repeats of a highly active synthetic auxin-response element and can reflect the in vivo auxin level [41]. The roots were subjected to GUS staining. Stained plant tissues were photographed using a stereomicroscope (Stemi 508; Zeiss, Gottingen, Germany) equipped with a colour CCD camera. All experiments included eight replicates.

### 4.5. qRT-PCR Analysis

Total RNA was isolated from the roots of rice plants under ${\mathrm{NH}}_{4}^{+}$ or ${\mathrm{NO}}_{3}^{-}$ supply. The RNA extraction, reverse transcription, and quantitative reverse transcription-polymerase chain reaction (qRT-PCR) methods were as previously described [42]. All experiments were with three replicates. The primer sets for *ARF**1* and *PIN2* are listed in Appendix A.

### 4.6. Data Analysis

Data were pooled to calculate means and standard errors (SEs) and subjected to one-way analysis of variance (ANOVA), followed by a Ryan–Einot–Gabriel–Welch F-test at *p* < 0.05 to determine the statistical significance of differences between treatments. All statistical evaluations were conducted using SPSS (version 11.0) statistical software (SPSS Inc., Chicago, IL, USA). All experiments included three independent biological replicates.

## 5. Conclusions

${\mathrm{NO}}_{3}^{-}$ stimulated LR formation within 7 d, but the stimulatory effect disappeared after 7 d, in parallel with the auxin response and transport in roots. *ARF1* was involved in the short-term ${\mathrm{NO}}_{3}^{-}$-induced LR formation. SL production was increased under ${\mathrm{NO}}_{3}^{-}$ treatment. The application of SLs and ${\mathrm{NO}}_{3}^{-}$ inhibited *PIN2* transcription. *PIN2* mutation inhibited the sensitivity of the response of LR formation to ${\mathrm{NO}}_{3}^{-\;}$application. These results demonstrate that ${\mathrm{NO}}_{3}^{-}$ modulated LR formation by affecting the auxin response and transport in rice roots, with SL involvement in the long term.

## Figures and Tables

**Figure 1 genes-12-00850-f001:**
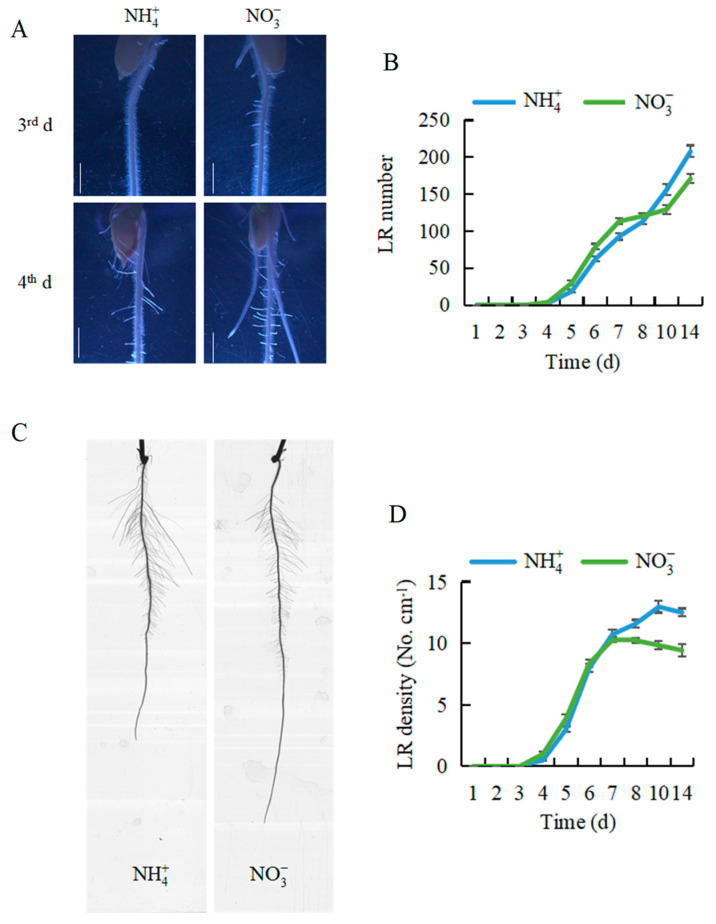
The lateral root (LR) density and LR number in wild-type (WT) plants. Seedlings were grown in a hydroponic media containing ${\mathrm{NH}}_{4}^{+}$ and ${\mathrm{NO}}_{3}^{-}$ for 14 d. (**A**) LR formation at 3rd and 4th d; (**B**) LR number over time; (**C**) The lateral root morphology in seminal root of the rice plants at 10 d; (**D**) LR density over time. Data are means ± SE, and bars with different letters indicate significant difference at *p* < 0.05 tested with ANOVA. d = days.

**Figure 2 genes-12-00850-f002:**
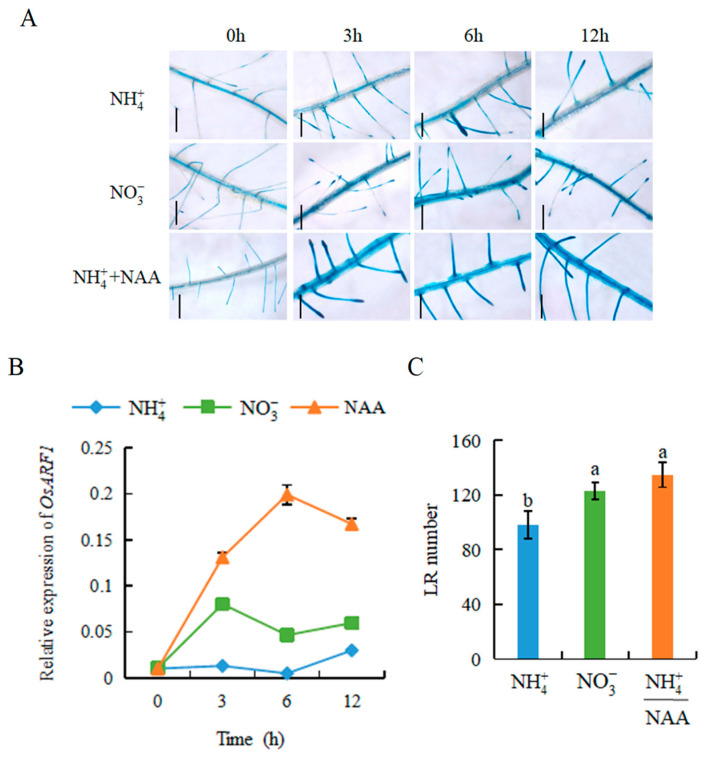
Histochemical localization of *DR5::GUS* and qRT-PCR analysis of *ARF1* gene, lateral root (LR) number in rice plants. Rice seedlings were grown in hydroponic media containing ${\mathrm{NH}}_{4}^{+}$, ${\mathrm{NO}}_{3}^{-}$, and ${\mathrm{NH}}_{4}^{+}$ +NAA treatments. Bar = 1 mm. (**A**) *DR5::GUS* in LR region; (**B**) Relative expression of *ARF1* over time; (**C**) LR number. Data are means ± SE, and bars with different letters indicate significant difference between treatments at *p* < 0.05. h = hours.

**Figure 3 genes-12-00850-f003:**
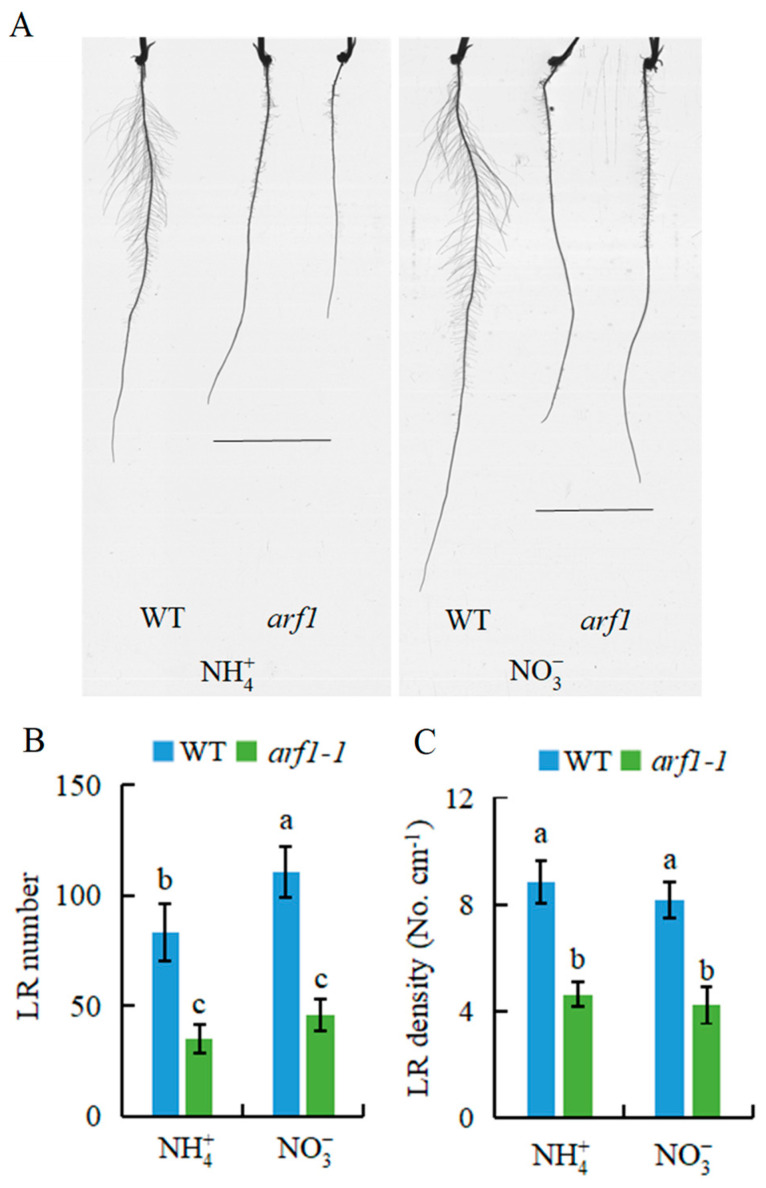
The lateral root (LR) number and LR density in *arf1* mutant plants. Seedlings were grown in a hydroponic media containing ${\mathrm{NH}}_{4}^{+}$ and ${\mathrm{NO}}_{3}^{-}$ conditions for 7 d. (**A**) The lateral root morphology in seminal root of the rice plants. (**B**) LR number; (**C**) LR density. Data are means ± SE, and bars with different letters indicate significant difference between treatments at *p* < 0.05. d = days.

**Figure 4 genes-12-00850-f004:**
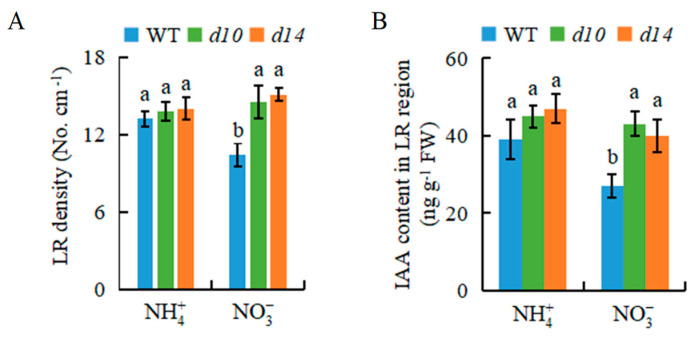
IAA content and LR density in wild-type (WT), *d10* (SL biosynthesis mutant), and *d14* (SL-responsive mutant) plants. Rice seedlings were grown in hydroponic media containing ${\mathrm{NH}}_{4}^{+}$ and ${\mathrm{NO}}_{3}^{-}$ for 10 d. (**A**) LR density; (**B**) IAA content in LR region. Data are means ± SE, and bars with different letters indicate significant difference between treatments at *p* < 0.05. d = days.

**Figure 5 genes-12-00850-f005:**
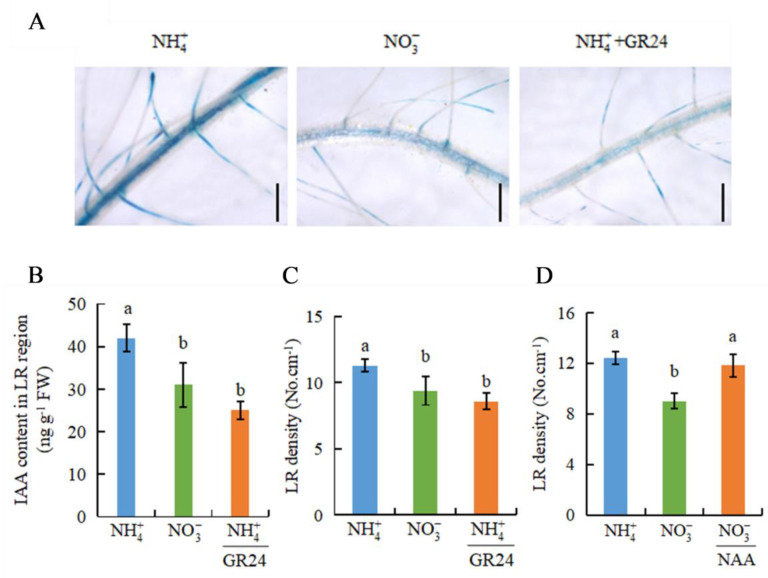
Histochemical localization of *DR5::GUS*, IAA content, and LR density in rice plants. Rice seedlings were grown in hydroponic media containing ${\mathrm{NH}}_{4}^{+}$, ${\mathrm{NO}}_{3}^{-}$, and ${\mathrm{NH}}_{4}^{+}$ +GR24 for 10 d. (**A**) *DR5::GUS* in LR region; (**B**) IAA content in LR region; (**C**,**D**) LR density. (**A**) Bar = 1 mm. Data are means ± SE, and bars with different letters indicate significant difference between treatments at *p* < 0.05. d = days.

**Figure 6 genes-12-00850-f006:**
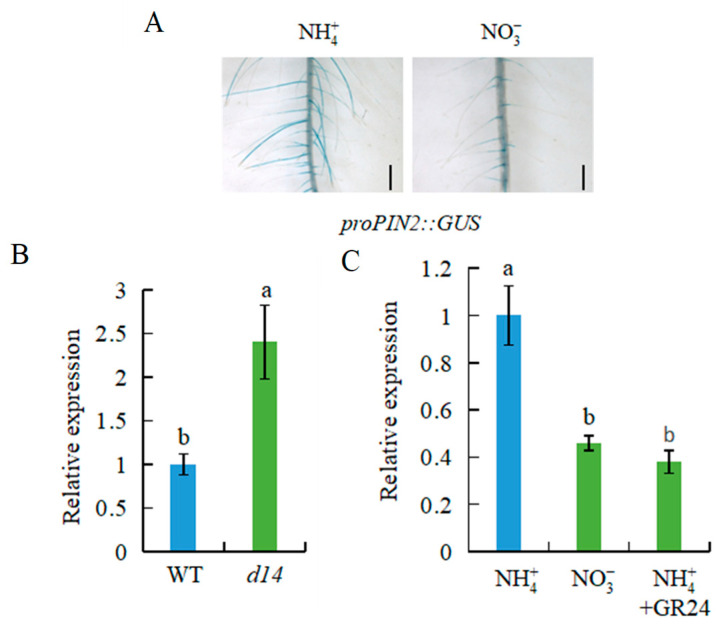
Histochemical localization of *proPIN2::GUS* and *OsPIN2* expression in rice plants. Rice seedlings were grown in hydroponic media containing ${\mathrm{NH}}_{4}^{+}$, ${\mathrm{NO}}_{3}^{-}$, and ${\mathrm{NH}}_{4}^{+}$ + GR24 for 10 d. (**A**) Localization of *pro**PIN2::GUS*; Bar = 0.5 mm. (**B**) *Os**PIN2* expression in WT and *d14* mutant under ${\mathrm{NH}}_{4}^{+}$ and ${\mathrm{NO}}_{3}^{-}$; (**C**) *Os**PIN2* expression in WT under ${\mathrm{NH}}_{4}^{+}$, ${\mathrm{NO}}_{3}^{-}$, and ${\mathrm{NH}}_{4}^{+}$ +GR24 conditions. Data are means ± SE, and bars with different letters indicate significant difference between treatments at *p* < 0.05.

**Figure 7 genes-12-00850-f007:**
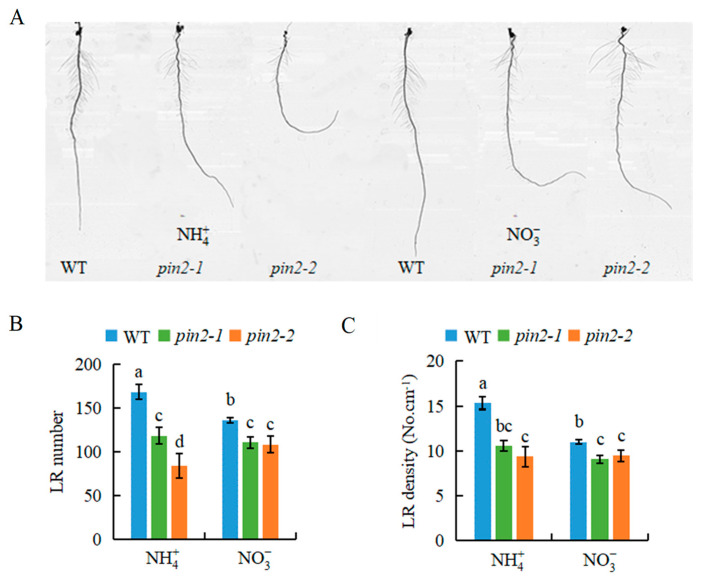
The lateral root (LR) number and LR density in *pin2* mutant plants. Seedlings were grown in a hydroponic media containing ${\mathrm{NH}}_{4}^{+}$ and ${\mathrm{NO}}_{3}^{-}$ for 10 d. (**A**) The lateral root morphology in seminal root of the rice plants. (**B**) LR number; (**C**) LR density. Data are means ± SE, and bars with different letters indicate significant difference between treatments at *p* < 0.05. d = days.

## Data Availability

Not applicable.

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
