# Peer review of "Nitrate Modulates Lateral Root Formation by Regulating the Auxin Response and Transport in Rice"

_genes, 2021, doi:10.3390/genes12060850_

Round 1
Reviewer 1 Report
I carefully read the paper “Nitrate-modulated lateral root formation by regulating auxin 2 response and transport in rice.”
The paper poses a very interesting focus, as the authors investigated relationships between NO3 supply and lateral root formation in rice and explored the involvement of auxins and strigolactones in the process.
Authors already investigated some correlated aspects of this pathway in the cited reference 7 (SPL14/17 act downstream of strigolactone signaling to modulate root elongation in response to nitrate supply in rice).
Methods are clearly described, and results are completely described. Discussion is essential but complete.
I have the main following concerns:
fig 5 is not visible, so results were considered as described in the tex.
At page 4 line 20, maybe the word density should be erased
Page 5 line 43, the fact that exogenous application of 37 SL affected IAA levels and LR formation did not demonstrate that NO3--inhibited LR formation by decreasing auxin 43 levels in roots at long term, with the involvement of SLs, but enforces this hypothesis.
In fig 6, b and d are related to wt or mutant? Why din the author not compare all the data in a sigle panel?
In fig 7 is NIP the same as wt?
In methods, PDR5:GUS should be better described
In general, language should be checked. In particular, I hardly found the verb in page 1 line 16, page 2 lines10-12, page 2 line 35, page 5 lines 4-6, page 8 lines 24 and 27.
Finally, letters C and D in line 29 at page 5 should be checked.
Author Response
I carefully read the paper “Nitrate-modulated lateral root formation by regulating auxin 2 response and transport in rice.”
The paper poses a very interesting focus, as the authors investigated relationships between NO3 supply and lateral root formation in rice and explored the involvement of auxins and strigolactones in the process.
Authors already investigated some correlated aspects of this pathway in the cited reference 7 (SPL14/17 act downstream of strigolactone signaling to modulate root elongation in response to nitrate supply in rice).
Methods are clearly described, and results are completely described. Discussion is essential but complete.
Answer: Thanks for giving valuable suggestion to help us to improve the manuscript. We revised our manuscript according to your comments.
I have the main following concerns:
fig 5 is not visible, so results were considered as described in the tex.
Answer: Thank you for your correction. We replaced it with a new Figure 5.
At page 4 line 20, maybe the word density should be erased
Answer: Thank you for your correction. We deleted the “density”.
Page 5 line 43, the fact that exogenous application of SL affected IAA levels and LR formation did not demonstrate that NO3--inhibited LR formation by decreasing auxin levels in roots at long term, with the involvement of SLs, but enforces this hypothesis.
Answer: Thank you for your correction. We revised the sentence to “These results showed that NO3--inhibited LR formation probably by decreasing auxin levels in roots at long term, and SLs may be involved in the process.”
In fig 6, b and d are related to wt or mutant? Why din the author not compare all the data in a sigle panel?
Answer: Thank you for suggestion.The b and d are WT plamts. We the compare b and d in a sigle panel .
In fig 7 is NIP the same as wt?
Answer: Thank you for your quesion. The NIP (Nipponbare ecotype) is WT, we revised the “NIP” to “WT”.
In methods, PDR5:GUS should be better described
Answer: Thank you for your suggestion. We added the relevant description about pDR5:GUS.
In general, language should be checked. In particular, I hardly found the verb in page 1 line 16, page 2 lines10-12, page 2 line 35, page 5 lines 4-6, page 8 lines 24 and 27.
Finally, letters C and D in line 29 at page 5 should be checked.
Answer: Thank you for your correction. We revised these sentences.

Reviewer 2 Report
The work could to be interested, however.
The Introduction should be condensed and shorted. Authors have to justify more the aims of research.
The many explanation is needed in Methods. The parts of this paragraph are unclearly and another than their subtitle. The research design is unclear.
The part ,,Plant growth” need to be described more clearly. It’s possible, the table would be made and will be more suitable.
The IAA analyze content should be better described, including HPLC chromatography. Which analyze were conducted? qualitative and quantitative analysis?
The results should be described more interesting. It seems to be poor in this version.
Numbering of the whole work should be corrected.
The some figures are unreadable.
The Authors have to formulate more conclusions.
Line 1 / 44-45: should be at the end.
Author Response
Thanks for giving valuable suggestion to help us to improve the manuscript. We revised our manuscript according to your comments.
The Introduction should be condensed and shorted. Authors have to justify more the aims of research.
Answer: Thank you for your suggestion. We condensed and shorted the Introduction.
The many explanation is needed in Methods. The parts of this paragraph are unclearly and another than their subtitle. The research design is unclear.
Answer: Thank you for your suggestion. We revised the Methods.
The part ,,Plant growth” need to be described more clearly. It’s possible, the table would be made and will be more suitable.
Answer: Thank you for your suggestion. We revised the “Plant growth” and added the related reference.
The IAA analyze content should be better described, including HPLC chromatography. Which analyze were conducted? qualitative and quantitative analysis?
Answer: Thank you for your suggestion. We revised the methods of determination of IAA content.
The results should be described more interesting. It seems to be poor in this version.
Answer: Thank you for your suggestion. We revised the results.
Numbering of the whole work should be corrected.
The some figures are unreadable.
Answer: Thank you for your correction. We replaced them with new Figures.
The Authors have to formulate more conclusions.
Answer: Thank you for your suggestion. We added more conclusions.
Line 1 / 44-45: should be at the end.
Answer: Thank you for your suggestion. We removed the sentence to Page 2 line 25.

Round 2
Reviewer 2 Report
The Figures (1-7) should be magnified because the roots and the font is unreadable and invisible, eg. NO3-. The punctuation also should be correct.
Page 2, line 15: Arabidopsis – italics
Page 2, line 34: [32-34] – correct please;
Page 2, line 42: ,,In this study, Tto further study”… – correct
Page 2, line 48: correct the numbering of paragraphs
Page 2, line 51, 54: Figures, and in whole work if needed.
Page 3, Line 33 – Page 4 Line 1-3 – incomprehensible
Page 5, Line 13-15: the sentences with the literature cited should be in the Discussion, not in Results
Page 8, Line 15: numbering
Page 8, Line 32: I think it would be better the discussion should be a one part
Page 9, Line 44: Numbering
Author Response
The Figures (1-7) should be magnified because the roots and the font is unreadable and invisible, eg. NO3-. The punctuation also should be correct.
Answer: Thank you for your correction. We revised these contents.
Page 2, line 15: Arabidopsis – italics
Answer: Thank you for your correction. We revised it.
Page 2, line 34: [32-34] – correct please;
Answer: Thank you for your correction. We revised the sentence.
Page 2, line 42: ,,In this study, Tto further study”… – correct
Answer: Thank you for your correction. We revised the sentence.
Page 2, line 48: correct the numbering of paragraphs
Answer: Thank you for your correction. We corrected the numbering of paragraphs.
Page 2, line 51, 54: Figures, and in whole work if needed.
Answer: Thank you for your correction. We revised it.
Page 3, Line 33 – Page 4 Line 1-3 – incomprehensible
Answer: Thank you for your correction. We revised it.
Page 5, Line 13-15: the sentences with the literature cited should be in the Discussion, not in Results
Answer: Thank you for your correction. We deleted the literature cited.
Page 8, Line 15: numbering
Answer: Thank you for your correction. We revised it.
Page 8, Line 32: I think it would be better the discussion should be a one part
Answer: Thank you for your correction. We revised the discussion to be a one part.
Page 9, Line 44: Numbering
Answer: Thank you for your correction. We revised it.
